# The Contribution of Gut Microbiota and Endothelial Dysfunction in the Development of Arterial Hypertension in Animal Models and in Humans

**DOI:** 10.3390/ijms23073698

**Published:** 2022-03-28

**Authors:** Jessica Maiuolo, Cristina Carresi, Micaela Gliozzi, Rocco Mollace, Federica Scarano, Miriam Scicchitano, Roberta Macrì, Saverio Nucera, Francesca Bosco, Francesca Oppedisano, Stefano Ruga, Anna Rita Coppoletta, Lorenza Guarnieri, Antonio Cardamone, Irene Bava, Vincenzo Musolino, Sara Paone, Ernesto Palma, Vincenzo Mollace

**Affiliations:** 1Laboratory of Pharmaceutical Biology, in IRC-FSH Center, Department of Health Sciences, University “Magna Græcia” of Catanzaro Italy, 88021 Catanzaro, Italy; v.musolino@unicz.it; 2IRC-FSH Department of Health Sciences, University “Magna Græcia” of Catanzaro Italy, 88021 Catanzaro, Italy; carresi@unicz.it (C.C.); rocco.mollace@gmail.com (R.M.); federicascar87@gmail.com (F.S.); miriam.scicchitano@hotmail.it (M.S.); robertamacri85@gmail.com (R.M.); saverio.nucera@hotmail.it (S.N.); boscofrancesca.bf@libero.it (F.B.); oppedisanof@libero.it (F.O.); rugast1@gmail.com (S.R.); annarita.coppoletta@libero.it (A.R.C.); lorenzacz808@gmail.com (L.G.); tony.c@outlook.it (A.C.); irenebava@libero.it (I.B.); palma@unicz.it (E.P.); mollace@libero.it (V.M.); 3Nutramed S.c.a.r.l, Complesso Ninì Barbieri, Roccelletta di Borgia, 88021 Catanzaro, Italy; sara.paone06@gmail.com; 4IRCCS San Raffaele, Via di Valcannuta 247, 00133 Rome, Italy

**Keywords:** blood pressure, hypertension, cardiovascular diseases, endothelium dysfunction, microbiota, intestinal dysbiosis

## Abstract

The maintenance of the physiological values of blood pressure is closely related to unchangeable factors (genetic predisposition or pathological alterations) but also to modifiable factors (dietary fat and salt, sedentary lifestyle, overweight, inappropriate combinations of drugs, alcohol abuse, smoking and use of psychogenic substances). Hypertension is usually characterized by the presence of a chronic increase in systemic blood pressure above the threshold value and is an important risk factor for cardiovascular disease, including myocardial infarction, stroke, micro- and macro-vascular diseases. Hypertension is closely related to functional changes in the endothelium, such as an altered production of vasoconstrictive and vasodilator substances, which lead to an increase in vascular resistance. These alterations make the endothelial tissue unresponsive to autocrine and paracrine stimuli, initially determining an adaptive response, which over time lead to an increase in risk or disease. The gut microbiota is composed of a highly diverse bacterial population of approximately 10^14^ bacteria. A balanced intestinal microbiota preserves the digestive and absorbent functions of the intestine, protecting from pathogens and toxic metabolites in the circulation and reducing the onset of various diseases. The gut microbiota has been shown to produce unique metabolites potentially important in the generation of hypertension and endothelial dysfunction. This review highlights the close connection between hypertension, endothelial dysfunction and gut microbiota.

## 1. Introduction

Blood pressure (BP) is the pressure exerted by blood against the walls of blood vessels and depends on the amount of blood that pumps the heart and the resistances that oppose its free flow. Physiological BP values are approximately 115/75 mm Hg [1,2]. The maintenance of these values is closely related to lifestyle; in particular, some non-modifiable factors, including genetic predisposition or pathological alterations, can negatively influence BP. On the other hand, modifiable factors, such as dietary fat and salt intake, sedentary lifestyle, overweight, inappropriate combinations of drugs, alcohol abuse, smoking and the use of psychogenic substances, are fundamental for the alteration of the homeostatic condition and for the alteration of BP [3]. It is also important to highlight that some commonly taken medications (steroids, non-steroidal anti-inflammatory drugs, nasal decongestants, oral contraceptives and antidepressants) can result in BP elevations [4]. The American College of Cardiology, the American Heart Association and the European Society of Hypertension have established behavioral guidelines to ensure the maintenance of baseline BP values, based on non-pharmacological interventions such as intensifying physical activity with a program of defined aerobic exercises, limited salt and alcohol intake, weight loss, and use of the Dietary Approaches to Stop Hypertension (DASH) diet, with a high intake of fruits and vegetables and low in fat [5,6,7].

Accordingly, even in guidelines for the identification and management of systemic hypertension in animals, such as dogs and cats, factors such as age and sex are related to increased BP [8]. Indeed, an association has been found between aging, sex, systolic and diastolic BP and mean arterial blood pressure values [9]. Among several factors (temperament, exercise regimen, breed or sex), age accounts for the majority of the variation in systolic and diastolic blood pressure in healthy dogs. In fact, a significant difference was found between all these parameters among young and elderly dogs, with an increase in systolic and diastolic blood pressure per year from 1 to 16 years of age of 1–3 mmHg [10]. Interestingly, as in humans with secondary hypertension, it has been suggested that increased blood pressure can be expected in dogs and cats with kidney disease, hypothyroidism, diabetes mellitus, liver disease and hyper-adreno-corticism. For the latter, a close relationship with increased blood pressure was found along with the risk of consequences such as retinal or renovascular damage [8,11].

Furthermore, obesity is an additional factor related to the increase in BP. Indeed, overweight animals have higher BP than normal weight ones [10]. A significant increase in blood pressure was observed in rabbits fed a high-fat diet compared to lean controls, and this finding was associated with higher glycosaminoglycan content in the kidney leading to impaired renal fluid regulation with the onset of arterial hypertension [11]. Moreover, it has been suggested that the presence of co-morbidities, such as chronic kidney disease, endocrinopathies or cardiopathies could explain the relationship between obesity and hypertension [12].

Hypertension is usually characterized by a chronic increase in systemic BP above a certain threshold value and is an important risk factor for the onset of cardiovascular diseases including myocardial infarction, stroke, micro- and macro-vascular diseases [13]. In fact, the progression of hypertension is associated with functional and structural cardiovascular abnormalities with damage to the heart, vessels, kidneys, brain and other organs. In humans, hypertension can cause alterations to other organs that will produce some typical symptoms; in fact, although the conduction arteries try to protect the tissues from large pressure excursions, some organs will be damaged. In particular, the heart will have to work with a greater load leading to ischemia [14], thickening of the arteries [15], rupture of the brain thin vessels generating brain hemorrhages [16], damage to the retinal arteries of the ocular fundus [17] and possible renal microangiopathy which will contribute to systemic arterial hypertension [18]. The main organ damage generated by hypertension, is shown in Figure 1.

Hypertension is often associated with renal damage, and it still represents one of the most important causes of end-stage renal disease. Kidney damage induced by hypertension determines morphological and functional renal alterations, which results in nephropathy associated to glomerular, tubular and interstitial injury. The severity of disease depends on several factors, such as the causes of disease, the presence of underlying kidney disease or the hypertension degree, as well as on individual susceptibility. In particular, it is well known that, among the complex mechanism involved in the mechanism of hypertension, kidney modifications occurs [19]. As is known, alteration of physiological pressure levels is linked to the modification of several neurohumoral system such as the role of natriuretic peptides and the endothelium, the sympathetic nervous system (SNS) and the immune system and in particular, the renin–angiotensin–aldosterone system (RAAS), alterations in which lead to increase in mean blood pressure, blood pressure variability and organ damage such as chronic kidney disease, associated with cardiovascular disease [20]. RAAS has the important role of balancing pressure and volume in the kidney in response to a reduction of extracellular fluid linked to sodium excretion. These actions are made by the ability of RAAS to regulate blood pressure as it mediates sodium retention and natriuretic pressure, salt sensitivity, vasoconstriction, endothelial dysfunction and vascular injury [20]. As for humans, and also in small animals, renal disease is a frequent event associated with hypertension. Similarly, there are multi-factorial causes leading to the development of renal hypertension, such as the above-mentioned sodium retention, or RAAS activation that could lead to kidney disease. Kidney disease is commonly associated with hypertension in dogs, cats and other species. There are multiple mechanisms underlying the development of renal hypertension including sodium retention, activation of the renin–angiotensin system and sympathetic nerve stimulation. The relative importance of these and other mechanisms may vary, both between species and according to the type of kidney disease that is present. Consideration of underlying disease mechanisms may aid in the rational choice of therapy in hypertensive patients performed by the gut microbiota [21]. Indeed, thanks to its metabolites, the gut microbiota stimulates the enteric afferent sensory fibers or affects the target organs responsible for BP regulation, such the kidney, as has been observed in the link between Gut’s SCFAs production and blood pressure alteration, which is due to olfactory receptors expressed in the vasculature and kidneys [22,23].

As with humans, a sustained increase in BP also causes damages to various tissues in animals. Studies in dogs, cats and humans have observed a link between proteinuria and renal damage [8]. Indeed, dogs with renal damage and hypertension have higher kidney lesion scores for mesangial matrix, tubule damage, fibrosis and cell infiltrate [24].

Moreover, in dogs with proteinuria associated with chronic renal glomerular damage, renal pathological analysis showed tubulointerstitial damage, fibrosis, atrophy, and renal inflammation [25,26].

Other lesions also affect the eye, due to hypertensive retinopathy with blindness, retinal detachment, and retinal hemorrhage with ocular lesion [27]; the brain, with encephalopathy associated with microhemorrhages and thrombi [28,29,30]; and the heart, with cardiovascular changes along with left ventricular hypertrophy, impaired cardiac function, increased heart rate, left ventricular mass index, decreased E/A ratio and diastolic dysfunction [31]. Histological observation showed multifocal dissolution and fragmentation of the myofilaments in cardiomyocytes with mitochondrial and cellular swelling, and loss of mitochondria and cells, along with myocardial fibrosis [32]. Although hypertension affects 30% of the population and 70% of the elderly, as well as being the most common cardiovascular risk factor, the etiology of most cases remains undefined [33]. In fact, in about 90% of cases the precise causes are unknown [34]. The functioning of the endothelium, the inner layer of blood vessels that controls circulation through the production of vasoactive substances and maintains vascular homeostasis, is one of the first targets in cardiovascular risk factors [35,36]. One of the first diseases associated with the reduced bioavailability of factors released by the endothelium is arterial hypertension, but, from the knowledge available in the scientific literature, it is not possible to determine whether endothelial dysfunction is the cause or a consequence of arterial hypertension [37]. Furthermore, a great deal of evidence describes the role of the gut microbiota in cardiovascular diseases, with particular attention to hypertension. In this review the role of the endothelium and intestinal microbiota in the phenomenon of arterial hypertension will be investigated in order to provide new therapeutic targets in the study, treatment and resolution of this pathology, which considerably increases cardiotoxic risk.

### 1.1. Endothelial Function and Dysfunction

Anatomically, the endothelium is made up of a monolayer of endothelial cells (EC) that extends along the entire circulatory system and forms the inner lining of blood vessels. Initially it was thought that the endothelium was made up of inert cells, with an exclusively structural function. To date, it is known that these cells are metabolically active, dealing with different physiological functions including the control of vasomotor tone, blood cellular traffic, maintenance of vascular homeostasis, permeability, proliferation, survival, innate and adaptive immunity. The main functions of the endothelium are the maintenance of vascular tone, cell adhesion, platelet aggregation, leukocyte trafficking, coagulation cascade, inflammation, permeability and regulation of thrombosis and fibrinolysis [38]. Vascular tone, defined as the balance between the degree of constriction of the blood vessel and its maximum dilation, is modulated by the release of relaxing and constricting factors derived from the endothelium. In fact, physiologically, ECs synthesize and release several endothelium-derived relaxing factors including vasodilator nitric oxide (NO), prostaglandins and endothelium-dependent hyperpolarization factors (EDHs). 

NO is a soluble gas that demonstrates important protective vaso-relaxing functions and is regulated by endothelial nitric oxide synthase (e-NOS), an enzymatic isoform constitutively expressed in ECs. This enzyme catalyzes the conversion of L-arginine into L-citrulline and NO. When NO is synthesized, it spreads into smooth vascular muscle cells, stimulating soluble guanyl cyclase and increasing cyclic guanosine monophosphate, (NO effector), which promotes vasodilation [39]. Nevertheless, the endothelium is also able to produce contracting factors such as endothelin, thromboxane A2, angiotensin II, and superoxide; thus the correct balance between the production of vasodilators and vasoconstrictors guarantees a correct maintenance of the vascular tone [40]. Endothelial cells are also able to relax vascular smooth muscle cells by generating EDH, the mechanism of which may vary depending on vascular beds and species. In fact, in some vascular beds and under specific conditions, endothelial cells generate EDH, release the calcium ion, presumably from the endoplasmic reticulum, and activate K+ channels. Their spread to adjacent smooth muscle cells, via myoendothelial gap junctions, induces endothelial-dependent hyperpolarization and relaxation in arteries’ resistance [41]. This mechanism makes it clear that EDH impairment can lead to both endothelial dysfunction and altered blood pressure regulation [42]. 

The endothelium also plays an important role of acting as a selective barrier between the blood and surrounding tissue, regulating the exchange of water, solutes and cells, and maintaining normal homeostasis. An alteration of this function can lead to hyper-permeability with the passage of blood components through the endothelium [43]. The passage of water and solutes occurs through the endothelial cell itself, from the luminal membrane to the basolateral membrane or in the opposite direction, through the so-called transcellular transport. Macromolecules cross the endothelium using the nearby endothelial transcellular space, connected by protein complexes, also called junctions. Cell junctions can be of three types: gap junctions, adherent junctions and tight junctions [44,45]. The endothelium plays a crucial role in maintaining blood fluidity and preventing thrombosis: in physiological conditions, it provides the correct hemostatic balance through different anticoagulant and antiplatelet mechanisms [46,47]. The endothelium is also involved in the formation of blood vessels. Indeed, endothelial cells produce a specific growth factor for vascular endothelium (VEGF) and recent studies have described a pattern of vascular formation that begins with the formation of immature vessels, which subsequently undergo remodeling and maturation [48]. Some of the main functions of the endothelium are represented in Figure 2. The inability of the endothelium to maintain vascular homeostasis is defined as endothelial dysfunction, a state in which the endothelial cell phenotype is altered, vasodilation is reduced, and the proinflammatory and prothrombotic state and the accumulation of reactive species are increased. Endothelial impairment includes two consecutive moments: (a) activation of endothelial cells; (b) overt endothelial dysfunction. The activation of endothelial cells occurs following an insult that causes the phenotypic transformation of the cells from a physiological condition to a pro-inflammatory state [49]. The pro-inflammatory phenotype is characterized by an increased expression of chemokines, cytokines, and cell adhesion molecules (CAMs) that facilitate the recruitment and adhesion of circulating leukocytes on the vascular wall [50]. If the harmful stimulus is prolonged and/or repeated there is the pathological transformation of the endothelium into an overt endothelial dysfunction, characterized by a pro-coagulant state, increased reactive oxygen species (ROS) levels, a sustained inflammatory state and constriction of blood vessels [51]. The blood cells that first reach the inflammatory site are the neutrophils which, after being firmly attached to the CAMs’ components expressed by the endothelium (selectins and integrins), cross the endothelial barrier, and thus leaving it. The process by which leukocytes transmigrate across endothelial barriers consists of six sequential steps: (1) rolling; (2) activation of leukocytes and endothelium; (3) adhesion to endothelial cells; (4) crawling toward the endothelial junctions; (5) trans-endothelial migration and (6) diapedesis [52]. 

Endothelial dysfunction is associated with most cardiovascular diseases (such as chronic heart failure, peripheral vascular disease, coronary heart disease, hypertension, diabetes, chronic kidney failure) [53,54,55], but also with severe viral infections, neurological diseases, and metabolic diseases among others [56,57,58,59]. ROS are reactive oxygen intermediates, which are physiological by-products of cellular metabolism. At physiological concentrations, ROS are very useful for cellular homeostasis, acting as second messengers in the cellular signals’ transduction and predisposing toxicity reactions against bacterial infections [60]. Conversely, when the amount of ROS exceeds the antioxidant capabilities of the cell or when the antioxidant enzymes have a reduced activity, oxidative stress occurs. This condition is extremely dangerous as ROS can react with major biological macromolecules and modify them accordingly [61,62]. Cell membranes are particularly susceptible to oxidative damage caused by ROS and can undergo “lipid peroxidation”, a process in which ROS remove electrons from lipids and damage phospholipids. This alteration can also lead the cell to apoptotic death [63,64]. The accumulation of ROS is involved in the onset of several diseases, including cancer, metabolic diseases, such as diabetes and obesity, neurodegenerative disorders, lung diseases and kidney diseases [65,66,67,68]. It has recently been shown that there is a close correlation between ROS accumulation, increased inflammation, and endothelial dysfunction [69,70,71]. For example, the reduction in the bioavailability of NO may be due not only to the reduced expression of e-NOS protein, but also to an increase in the level of ROS and above all of anion superoxide (O^•−^), responsible for the formation of peroxy-nitrite (ONOO^−^). The latter promotes protein nitration contributing to endothelial dysfunction and cell death [72,73]. The pathologies in which the availability of NO is altered, and which determine a harmful impact on the endothelium, through the mechanism described are multiple and include not only metabolic disorders (hyperglycemia, hyperlipidemia, hypertension) [74,75], but also prolonged exposure to drugs, aging and mental disorders [76,77,78,79]. To date, it is known that oxidative stress favors the extravasation of leukocytes: indeed, some scientific studies have shown that the exposure of Human Umbilical Vein Endothelial Cells (HUVECs) with hydrogen peroxide for 1 h, involves the translocation of selectin P on the cell surface, thus increasing the adhesion of neutrophils [53,80]. At the same time, antioxidant cell treatment removes O^•−^ accumulation reducing the early stages of the extravasation mechanism [81,82]. The Intercellular Adhesion Molecule 1 (ICAM 1) is a protein continuously present at low concentrations in the membranes of leukocytes and endothelial cells, which when stimulated by cytokines, considerably increases its concentration. ICAM-1 is another inflammatory mediator that plays a key role in neutrophil recruitment and neutrophil-mediated vascular injury. In addition, ICAM-1 is regulated at the level of gene transcription by numerous transcription factors involved in oxidative stress, including NF-kB, and Activator protein 1 (AP-1), highlighting the involvement of ROS in the transcriptional activation of this adhesion molecule. Treatment with hydrogen peroxide has been shown to rapidly increase ICAM-1 mRNA and its cell-to-surface protein expression in HUVECs [82,83]. Furthermore, treatment with the antioxidant N-acetyl-L-cysteine (NAC) reduces the expression of ICAM-1 by reducing the extravasation of leukocytes [84]. The accumulation of ROS can facilitate the induction of the autophagic process, which determines the elimination of dysfunctional mitochondria. However, when autophagy is excessive, it can lead to cell death. Recent experiments have shown that antioxidant treatment reduces ROS excess, restores proper autophagy, and reduces endothelial cell death [57,85,86]. The inflammatory process is closely related to the accumulation of ROS in endothelial dysfunction. Indeed, these two effects are complementary to each other and it is difficult to determine which effect occurs first: they can certainly be considered the common denominator of endothelial dysfunction [87,88]. The endothelium plays an important role in the initiation of the inflammatory process. The first step involves, as already mentioned, endothelial cells undergoing an activation process, usually classified into two types: type I activation is rapid and generates a transient response in which endothelial cells interact with leukocytes and platelets after loss of their cellular junctions and subsequent release of the P-selectin adhesion protein; type II activation is slower but more persistent and affects the expression of a variety of proinflammatory cytokines, including tumor necrosis factor-α (TNF-α) and interleukin-1 (IL-1) [89]. Following the activation of endothelial cells, there is an increase in vascular permeability, the expression of pro-inflammatory cytokines, chemokines and enzymes, and an over-expression of adhesion molecules, such as ICAM-1, selectin-E and vascular cell adhesion molecule-1. This condition generates a real inflammatory process, responsible for the activation of leukocytes [90]. The release of inflammatory mediators, including vasoactive amines (histamine and serotonin), peptides (bradykinin, protease of thrombin) and eicosanoids (thromboxanes, leukotrienes and prostaglandins) and the recruitment of leukocytes continues with an intracellular inflow of calcium ion. Calcium is a key second messenger involved in the signaling pathways that affect endothelial permeability. The mediators produced bind their receptors on endothelial cells, triggering the opening of calcium channels; the increased concentration of this ion activates calcium-dependent proteins. Calcium influx is involved in a number of phases of the inflammatory cascade and the leukocyte extravasation cascade, including leukocyte rolling, adhesion, migration and diapedesis [91]. It is increasingly confirmed that hypertension reduces endothelium-dependent relaxation in both large and small arteries [92]. In general, most studies on endothelial dysfunction focus on mechanisms related to the reduced bioavailability of NO resulting from both decreased NO production and increased NO degradation. In hypertension, the mechanisms involved in changes in NO metabolism are decreased NO production and increased NO inactivation [93,94]. A reduced NO synthesis can result from:(a)a deficiency in the substrate of NO synthase, L-arginine [95];(b)a high concentration of endogenous NO synthase inhibitors [96];(c)deficiency of cofactor for NO endothelial synthesis [97];(d)reduced expression of e-NOS [98];(e)an alteration of the transduction signals leading to the uncoupling of endothelial NO synthase [99];

### 1.2. Endothelial Dysfunction and Hypertension

Considering these findings, it is known that hypertension is associated with an increase in ROS production and a decrease in the level of antioxidants [100,101]. The connection between oxidative stress and hypertension is also demonstrated by the knowledge of scientific data, which shows that vitamin C restores NO production and improves endothelial function in hypertension. In arterial hypertension, the reduction in the bioavailability of NO, also justified by its binding with various molecules such as hemoglobin or albumin, becomes more persistent due to the interaction of NO with O^•−^. In fact, in physiological conditions, the endogenous antioxidant system limits the interaction between NO and O^•−^. On the contrary, pathological diseases, such as diabetes mellitus, tobacco smoke, hyperlipidemia and hypertension, lead to an increase in the production of O^•−^, which results in a reduction of NO availability. In this case, NO reduction is responsible for vasoconstriction, arterial remodeling, arterial stiffness, thickening of the subendothelial sheet, increase in the amounts of proteins, lipids and proinflammatory cells, changes in the viscoelastic properties of the arterial wall, the onset of local inflammatory processes and increased leukocyte adhesion [102,103]: precisely for this reason, antioxidant agents are included in the treatment of hypertension. Hypertension is closely related to functional changes in the endothelium and hypertensive patients exhibit endothelial-associated vasodilation impairment with an abnormal NO function. The impaired production of vasoconstricting and vasodilating substances can lead to an increase in vascular resistance. These alterations make the endothelial tissue unresponsive to autocrine and paracrine stimuli, initially determining an adaptive response which over time leads to an increase in risk or disease. Therefore, targeting of endothelial dysfunction is crucial for the treatment of hypertensive subjects [104,105,106]. 

In rabbits, intravenous administration of a NOS inhibitor, N omega-monomethyl-L-arginine (L-NMMA), induced a dose-dependent increase in mean systemic arterial blood pressure. This increase in BP was due to inhibition of NO release from ex vivo perfused aortic segments. L-arginine infused through the aortic segments was able to reverse this inhibition within 15 min, demonstrating that L-arginine-derived NO in the vascular endothelium is important in regulating blood pressure [107]. Furthermore, L-arginine was also able to counteract the inhibitory effect on endothelium-dependent relaxation of L-NG-methylarginine, which after its infusion in guinea pigs leads to a dose-dependent increase in systolic and diastolic blood pressure [108].

Moreover, changes in NO-mediated vasodilatations via L-NMMA infusion in dogs were highlighted by a dose-related increase in aortic pressure, decreased rest phasic coronary blood flow and heart rate, and by the increase in basal epicardial coronary vasomotor tone [109].

In all in vivo models of hypertension there is a high impairment of endothelium-dependent vessel relaxation in both large and small resistance arteries. 

Endothelial dysfunction is also well described in human hypertension, in small and large epicardial coronary arteries [110,111]. Hypertension is not only responsible for changes in endothelial regulation of vasomotor function, but also for the induction of inflammation in the vascular wall [112]. Hypertension contributes to the endothelial development of many serum markers’ inflammation, including cytokines, interleukin-6, alpha tumor necrosis factor, CAMs, fibrinogen and C-reactive protein [113]. A representation of endothelial involvement in hypertension is shown in Figure 3.

## 2. Gut Microbiota and Hypertension 

The gut microbiota is composed of a very varied bacterial population of about 10^14^ bacteria and whose intestinal colonization begins at the birth of the organism. Although the fetal microbiota has been shown to depend on the maternal microbiota, it is now known that its composition differs within the first 3–5 years of life [114]. The microbiota of an adult individual is mainly characterized by three enterotypes, each of which is composed of a prevalent bacterial species: enterotype 1 is characterized by *Bacteroides*, which recover maximum energy from the fermentation of carbohydrates and proteins. They are also responsible for the production of biotin, riboflavin, pantothenic acid and ascorbic acid. Enterotype 1 is often related to a diet high in animal proteins and fats and low in fiber and plants, such as the Western diet. Enterotype 2 is characterized by a preponderance of bacteria of the genus *Prevotella*, which can degrade host glycans and proteins. They produce high levels of thiamine and folic acid. Enterotype 2 is related to a diet high in fiber and carbohydrates. Enterotype 3 is characterized by bacteria of the genus *Ruminococcus*, which can colonize the superficial mucosa, consume simple sugars, and could play an important role in modulating the immune system. Due to the absorption of carbohydrates and simple sugars, enterotype 3 can be related to a tendency to gain weight [115]. After birth, breastfeeding involves the intake of lactose, the main carbohydrate in human milk, which promotes the growth of *Lactobacillus*. Subsequently, the consumption of solid foods reshapes the intestinal microbiota, with the colonization by anaerobic *Proteobacteria, Lactobacillus* and *Bifidobacterium*. Subsequently, several genera will appear within the phylum *Bacteroidetes* [116,117]. In healthy adults, the composition of the gut microbiota is stable but diet and environmental factors can generate imbalances. In the elderly population there is a reduction in bacterial diversity, with an increase in some *Proteobacteria* and a decrease in *Bifidobacteria*. This modification causes an interruption of the intestinal barrier function and the likelihood of toxin accumulation [118]. The integration of probiotics and prebiotics has been shown to have protective effects on health, with a reduction in chronic inflammation and increase in longevity [119,120]. This important observation highlights the importance of the gut microbiota in maintaining general health [121]. Therefore, it is clear, that a balanced intestinal microbiota preserves the digestive and absorbent functions of the intestine, limits the invasion of pathogens and toxic metabolites into the circulation and reduces the occurrence of various diseases [122,123,124,125,126,127]. The gut microbiota has been shown to produce unique metabolites that are potentially important in blood pressure control. These bacteria are the only source of short-chain fatty acids (SCFAs), products of the digestion of dietary fibers and the fermentation of undigested carbohydrates, from the gut microbiota, for the body. SCFAs are fatty acids with less than six carbon atoms, including the most abundant, acetic acid, propionic acid, butyric acid, and the least abundant, valeric acid and caproic acid. Bacteria synthetize SCFAs in sequential steps from glycolysis of glucose to pyruvate, acetyl-CoA and finally acetic acid, propionic acid, and butyric acid. SCFAs are known to be beneficial metabolites for blood vessel control and can influence the immune, epithelial, nervous and blood vessel systems and modulate blood pressure, reducing the risk of hypertension [128]. Indeed, hypertension has been associated with a decreased intestinal microbial diversity and SCFA-producing bacteria [129]. Furthermore, the abundance of SCFA-producing bacteria in pregnant women is negatively correlated with blood pressure [130]. Mechanistically, SCFAs has been shown to bind to a G protein by activating intracellular signaling in various cell types [131] and by modulating blood pressure [132,133]. Indeed, a relationship was found between Gpr41, a G protein-coupled receptors, and SCFAs produced by the gut microbiota. In particular, the binding of SCFAs to Gpr41 leads to a stimulation of renin secretion which results in a BP increase. Moreover, Grp41 has been shown to influence the endothelium in determining vasodilation and that the absence of this G protein in KO mice results in isolated systolic hypertension compared to wild type mice, together with an elevated pulse wave velocity, as observed by the telemetry measurement. These effects suggest that the gut microbiota, through SCFAs, can influence BP regulation [134]. The modulation of BP, SCAF mediated, occurs also with many sensory receptors (olfactory receptors, taste receptors). These receptors, in addition to playing their role in sensory tissues, also act in other districts where they serve as selective and sensitive chemo-receptors. For example, olfactory receptors (OR) are expressed in a variety of tissues in mice, humans, and other primates, and their ligands are often generated by physiological or metabolic processes [135,136]. Olfactory Receptor78 (Olfr78) is expressed in olfactory epithelium, but also in renal afferent arteriole, the site where renin (the initial, limiting passage in the renin–angiotensin–aldosterone pathway) is stored for possible release into the blood, playing an important role in blood pressure control, autoregulation of tissue blood flow, and/or extracellular fluid volume regulation. In vitro and in vivo studies have shown that Olfr78 is a receptor for SCFAs, in particular acetate and propionate [137,138]. 

Moreover, SCFAs demonstrate an epigenetic effect on epithelial cells by acting on histone deacetylase [139] and by increasing the transcription of interleukin-10, an anti-inflammatory and immunosuppressive cytokine produced by a variety of mammalian cell types and capable of reducing inflammation, inhibiting the synthesis of pro-inflammatory cytokines such as IFN-γ, IL-2, IL-3, TNFα. Iterleukin-10 also exhibits a potent ability to suppress the antigen presenting-capability of antigen presenting-cells. Knockout studies have suggested that interleukin-10 acts as an essential immunoregulator in the intestinal tract: in fact, patients with Crohn’s disease react favorably to treatment with recombinant interleukin-10 producing bacteria [140,141]. 

The correlation between SCFA concentrations, inflammatory status and dysbiosis has also been demonstrated in dogs with chronic enteropathy which show lower fecal concentration and altered SCFA patterns associated with fecal microbiota modifications [142]. Dysbiosis caused by inflammatory bowel disease resulted in a reduction of SCFA-producing bacteria and alterations in ileal and colon mucosal bacteria in dogs, as observed by the increase in adherent bacteria, such as total bacteria, Enterobacteriaceae, E. coli and the presence of invasive bacteria, such as Enterobacteriaceae, E. coli, and Bacteroides in the sites of intestinal mucosa. Overall, these effects lead to worsening of the clinical disease [143,144].

SCFA maintains the epithelial barrier to reduce inflammation and their reduction has been identified as a responsible factor for the increase in blood pressure in obese pregnant women [145]. Overall, these findings support the assumption that the gut microbiota reduces the risk of hypertension [146,147,148]. However, it is important to note that some intestinal bacteria by-products have direct negative effects on blood pressure in the case of intestinal dysbiosis [149]. The composition of the gut microbiota is constantly changing and evolving and is influenced by several factors including diet, intestinal mucosa, drug consumption/abuse, the immune system, and the microbiota itself. When there is a correct balance between the composition of the gut microbiota and its entire genetic heritage (microbiome), a condition known as “intestinal eubiosis” is established. Conversely, when there are reductions in microbial diversity with expansion of specific bacterial taxa, a state of dysbiosis occurs [150]. Therefore, dysbiosis is a condition of microbial imbalance caused by an excessive growth of “harmful” bacteria inside the intestine, which cause irritation and predispose to the onset of various diseases including ulcerative colitis and Crohn’s disease, necrotizing enterocolitis, colorectal cancer, autoimmune diseases and neurological disorders [127,151,152,153,154,155,156]. The close connection between intestinal dysbiosis and hypertension has been well studied and highlighted [157]. Some by-products of intestinal bacteria can induce, as previously described, the onset of diseases in the case of intestinal dysbiosis and this can occur when a systemic inflammatory process is induced. For example, microbiota-derived metabolites can easily cross the blood-brain barrier (BBB), affecting the inflammatory state in the brain and inducing pathologies such as multiple sclerosis [158] and hypertension [159]. Probiotic and prebiotic supplementation should be administered in all adult/elderly individuals, to ensure intestinal eubiosis and also to normalize blood pressure. Figure 4 shows a diagram representing the protective role of the gut microbiota in a condition of eubiosis.

### 2.1. Gut Microbiota and Endothelial Dysfunction

The maintenance of endothelial function also occurs in relation to the endogenous mediators with which it comes into contact. Therefore, the derivatives of the metabolism of the microbiota can contribute to the physiology of the organism by helping to maintain homeostasis or causing the onset of diseases. It has also been shown that these metabolites can demonstrate different effects on endothelium function [160]. In particular, intestinal bacteria can affect the endothelium of the circulatory system through two main pathways: on the one hand, the microbiota and its metabolites can stimulate the enteric nervous system and, consequently, the activity of the brain centers that control the cardiovascular system; on the other hand, it can enter the bloodstream, through the blood–intestinal barrier, modulating the function of the tissues and organs that control the homeostasis of the circulatory system [161]. For this reason, the maintenance of the microbial composition in a state of eubiosis and the attenuation/resolution of intestinal dysbiosis has been proposed as a strategy to reduce endothelial and vascular dysfunction [162]. SCFAs, as previously mentioned, have beneficial effects on the endothelium and blood vessel control. Conversely, there are some harmful metabolites that will be investigated below. Trimethylamine (TMA), an organic compound with the formula N(CH_3_)_3_, is a tertiary amine produced in humans following the ingestion of foods from certain plants and animals, containing choline, phosphatidylcholine, glycerol-phosphocholine, carnitine, betaine, lecithin and L-carnitine. These substrates provide the gut microbiota with the ability to form TMA, which is absorbed into the bloodstream and subsequently oxidized to trimethylamine N-oxide (TMAO) in the liver [163]. TMAO is considered a cardiac risk biomarker as it possesses pro-atherogenic properties and is capable of predicting myocardial infarction, stroke or death [164]. In particular, the mechanism of action involves endothelial dysfunction and the prothrombotic effect caused by platelet aggregation [165]. TMAO-induced endothelial dysfunction occurs following activation of the transcription factor NF-kB, responsible for up-regulation of inflammatory signals and adhesion of leukocytes to endothelial cells [166,167]. In addition, in vitro studies have shown that high plasma TMAO levels were related to a reduction in circulating endothelial metabolites, increased endothelial dysfunction and severe cardiovascular events [168,169]. Moreover, it was found that an increase in TMAO is related to endothelial dysfunction and atherosclerosis [170] and that mice with high TMAO values, following a diet rich in choline, showed high endothelial damage, evident dyslipidemia and hyperglycemia [171]. Furthermore, an interesting clinical study showed that a high plasma level of TMAO was associated with increased inflammation and concomitant reduction of endothelial progenitor cells in patients with cardiovascular disorders [172]. In addition to the mechanisms already described, TMAO downregulates the expression of the anti-inflammatory cytokine IL-10, which can protect the endothelium from damage caused by increased inflammation and oxidative stress [173] TMAO leads to ROS generation and reduction of nitric oxide, both of which exert adverse effects in maintaining normal vascular function [174]. In an important and elegant study, Matsumoto et al. highlighted another mechanism by which TMAO is able to alter the vascular endothelial function. In fact, the effects of TMAO on endothelial-dependent relaxation in two arteries, in particular the upper mesenteric artery and the femoral arteries, have been studied and it has been shown that TMAO is able to induce both the inhibition of EDH and its consequent arterial relaxation. However, it is important to note that the described phenomenon does not occur in all vascular beds, but acts selectively: in this case, in particular, the femoral arteries are involved, but not the upper mesenteric artery [175].

A relationship between blood levels of TMAO, increased risk of mortality and renal insufficiency was found in humans and animals as observed by renal tubulointerstitial fibrosis, with increased levels of the early renal injury marker KIM-1 and enhanced phosphorylation of Smad3, and renal dysfunction observed by elevated cystatin C values after choline intake [176].

Finally, TMAO impairs the self-healing ability of damaged endothelial cells, leading to irreversible endothelial dysfunction [177]. 

Uremic toxins are metabolites derived from the metabolism of the gut microbiota of amino acids that contain aromatic groups, such as tyrosine, phenylalanine and tryptophan. The gut microbiota metabolizes these amino acids in the host liver to produce certain toxins, such as indoxyl sulfate, indoxyl glucuronide, indoleacetic acid, p-cresyl sulfate, p-cresyl glucuronide, phenyl sulfate, phenyl glucuronide, phenylacetic acid, and hippuric acid [178,179]. These circulating nitrogen metabolites are considered to be a predictive biomarker of coronary atherosclerosis [180]. Uremic toxins alter endothelial balance and promote dysfunction by activating NF-kB transcription factor signalling, overriding ICAM-1, the endothelial- and leukocyte-associated transmembrane protein long known for its importance in stabilizing cell–cell interactions, and monocyte chemoattractant protein-1 (MCP-1), which plays an important role in the selective recruitment of monocytes, neutrophils, and lymphocytes [181]. Furthermore, these toxins inhibit NO synthesis and increase ROS accumulation. The involvement of oxidative stress is provided by the consideration that the antioxidants N-acetylene and apocynin can mitigate the pro-apoptotic effect of p-cresyl sulfate in the endothelium [182]. In addition, the treatment with caffeic acid, a polyphenol present in white wine with antioxidant properties, was able to restore NO production and reduce ROS [183,184]. 3-hydroxyphenylacetic acid (3-HPAA), other metabolites produced by the gut microbiota after the intake of polyphenol-rich foods, and in particular quercetin have been shown to be potentially beneficial in hypertension. Indeed, in spontaneous hypertensive rats a dose-dependent reduction in mean systolic and diastolic pressure, associated with no heart rate changes, was observed after administration of 3-HPAA, by bolus or after slow intravenous infusions, but not by intravenous injection, suggesting that this effect was based only on peripheral relaxation. Meanwhile, 3-HPAA treated porcine coronary arteries isolated from pigs’ hearts showed a dose dependent vasodilatory response mediated by endothelium-derived NO [185].

Finally, intestinal microorganisms release proteins and peptides that act not only on other bacteria, but also on the rest of the body. Pathological bacteria are able to release peptides that destroy the blood–intestinal barrier, resulting in the spread of bacteria in the bloodstream, a considerable increase in inflammatory state and induction of permeability, transmigration and angiogenesis in the intestinal microvascular endothelial cells [186]. 

Along with human observation, a reduction of fecal bacterial diversity has been observed in cats affected by chronic kidney disease, which is associated with a significantly higher blood indoxyl sulfate concentration [187,188]. Likewise, in cats and dogs with chronic kidney disease with persistent azotemia, it was shown that increased levels of indoxyl sulfate were related to serum phosphorous concentration, loss of renal function, and smaller kidneys compared to non-azotemic cats [189]. Moreover, the increased levels of uremic toxins were associated with the increased concentration of fibroblast growth factor-23 [190] and with the increase of blood urea nitrogen, serum creatinine phosphate and the decrease of hematocrit [191].

Although the study of the direct correlation between gut microbiota and the development and progression of hypertension in animals, such as dogs and cats, is still premature, some interesting noteworthy research has been conducted in relation to other known cardiovascular diseases.

For example, canine degenerative mitral valve disease (DMVD) is one of the most common forms of cardiovascular disease in the dog and shares several molecular and pathophysiological similarities with that of humans.

Recent studies have shown that high circulating concentrations of TMAO and its nutrient precursors, including choline and L-carnitine, phosphatidylcholine, betaine, and trimethyl-lysine, together with uremic toxins, such as guanidino compounds and urea, were recorded in dogs with DMVD and Congestive Heart Failure (CHF) compared to asymptomatic or healthy dogs [192,193]. Interestingly, some of the short-chain and long-chain acyl-carnitine concentrations were reduced after a targeted dietary intervention mainly based on medium-chain triglycerides, fish oil and antioxidants [194,195].

The main question for the authors is to establish whether the increased concentrations of TMAO and its precursors registered represent the cause of the development and progression of DMVD and CHF or an effect of these conditions. Elevated TMAO concentrations are the result of impaired cardiovascular energy metabolism, or alternatively may be related to the inflammation associated with cardiovascular diseases.

A pilot study recently published on Scientific Reports clearly showed that quantifiable dysbiosis occurs in dogs with CHF due to increased levels of *Proteobacteria*, with a particular increase in *Escherichia coli* and an unclassified species of *Enterobacteriaceae*, suggesting a similar pattern to that described in human patients [196,197]. In agreement with previous studies, the authors correlated the elevated levels of *Escherichia coli* with the increased concentrations of TMAO in dogs with CHF. Furthermore, they pointed out the opportunistic nature of these bacteria; indeed, while some strains of E. coli are benign, some other are compatible with pathobionts inducing inflammation and contributing to inappetence, malnutrition and cachexia seen in dogs with CHF [198].

An interesting study published on 2021 identified, for the first time the relationship between gut microbial dysbiosis and circulating gut-derived metabolites in dogs with preclinical mixomatous mitral valve disease (MMVD) or with CHF secondary to MMVD, compared to healthy dogs [199]. In particular, the authors showed greater alpha and beta diversities in the gut of healthy dogs than in the dogs with MMVD, identified changes in five genera and six species of bacteria and clearly demonstrated that the dysbiosis index progressively increased with the severity of MMVD. Moreover, the dysbiosis index was inversely associated with *Clostridium hiranosis,* a key bile acid converter in the gut, while secondary bile acids promote the growth of beneficial bacteria and inhibit that of harmful ones. Finally, a positive correlation was identified between the key intermediates of long-chain fatty acid transport and oxidation, circulating short-chain acyl-carnitines and gut bacteria Lactobacillus and Megamonas, whose levels are reduced in MMVD dogs [199]. 

Thanks to these and other studies, the so-called “gut hypothesis” has been confirmed. According to this hypothesis, gut dysbiosis arises in the preclinical stages of the disease when no symptoms of cardiac remodeling are detectable, laying the foundations for a future targeted diagnostic and therapeutic approach. The beneficial or harmful effects of circulating metabolites of the gut microbiota on endothelium are shown in Figure 5.

### 2.2. Gut Microbiota and LPS/TLR4 Signal Transduction

The correlation between microbiota and hypertension has been studied experimentally using numerous animal models including spontaneous hypertensive rats, Dahl-sensitive rats, angiotensin-II induced hypertensive rats and deoxycorticosterone acetate (DOCA)-salt mice [200,201,202,203]. The results obtained showed that hypertension is accompanied by marked differences in the composition of the microbiota and their metabolites. In particular, there is less abundance of SCFA-producing bacteria, less abundance of *Bacteroidetes*, more abundance of lactate-producing bacteria and more abundance of *proteobacteria* and *cyanobacteria* [204]. Hypertension has been associated with lower gut microbial alpha diversity in several cross-sectional studies; in fact a greater abundance of Gram-negative bacteria has been appreciated, such as *Klebsiella, Parabacteroides, Desulfovibrio* and *Prevotella* [23,205,206]. Gram-negative bacteria are a source of endotoxins, such as lipopolysaccharides (LPS), which are pro-inflammatory molecules. The potential mechanisms contributing to hypertension development linked to dysbiosis involve: (1) metabolism-dependent pathways, consisting in a decrease in SCFA and TMAO production; (2) metabolism-independent pathways: LPS and peptidoglycan translocation [207]. In animal studies, LPS has been commonly used to induce vascular dysfunction [208], while in human samples the presence of high levels of LPS in the bloodstream has been identified as “endotoxemia” and has been correlated with cardiovascular disease and mortality [209]. Lower gut microbial alpha diversity in hypertension leads to intestinal dysbiosis with impaired integrity of the intestinal barrier resulting in the entry of LPS into the blood stream. In this way, LPS can advance intestinal dysregulation creating positive feedback damage. Bacterial endotoxins are recognized by toll-like receptors (TLRs), 13 integral I-type transmembrane receptors that play an essential role in hypertension and produce low-grade chronic inflammation, vascular remodeling, and oxidative stress [210]. TLRs are known for their ability to recognize evolutionarily conserved components of microorganisms, including bacteria, viruses, fungi and parasites [211]. Among these, TLR4 is the most commonly explored in hypertension. TLR4 binds LPS with the help of LPS-binding proteins and contribution of the MD-2 protein, stably associated with the extracellular fragment of the receptor. Prolonged activation of TLR4 is associated with several human neurodegenerative and autoimmune diseases and cancer [212,213]. When bacterial LPS binds to TLR4, this Complex activates the nuclear factor kappa-light-chain-enhancer of activated B cells (NF-κB) and promotes the subsequent inflammasome activation [214]. The infammasome serves to promote autoproteolysis and activation of caspase-1, which, in turn, cleaves pro-IL-1β and pro-IL-18. In summary, animal studies suggest a causal link between gut microbiota composition and BP regulation. In fact, the use of prebiotics has determined the reduction of BP in hypertensive patients [215]. Despite these favorable outcomes, it is still not clear how TLR4 affects BP under normal and hypertensive conditions and additional and specific studies should be organized.

### 2.3. High Salt Intake, Hypertension and Gut Microbiota 

In order to maintain the balance of liquids and cellular homeostasis, the human body needs a very small amount of salt. Over time, however, salt consumption has increased exponentially both because of a diet based on the “emphasis of flavor” (diet developed in Western countries), and the development of food technologies that use salt as a preservative in many foods. The result has been a consumption of a quantity of salt that exceeds by approximately 20 times the real requirement [216]. Since the human body is not adapted to expel this large amount of salt, multiple repercussions on our health have occurred, motivating millions of deaths per year [217]. To date, it is known that excess salt in the diet is an important risk factor for hypertension and the onset of cardiovascular disease; for this reason, the American Heart Association has recommended the correct amount of salt to be taken [218]. The salt should not exceed 2300 mg per day, although less than 10% of the US population observes this recommendation [219]. In addition, large numbers of individuals are hypersensitive to salt changes and develop BP alterations even if they are normotensive subjects. An excess of salt involves organ damage in the kidney, vasculature, and central nervous system, although it has recently been discovered that even the intestinal microbiota and immune cells can perceive excesses of Na^+^ and contribute to inflammation and hypertension [220,221,222]. The involvement of the gut microbiota has been demonstrated with some experimental evidence: first of all the transplantation of the intestinal microbiome of hypertensive subjects causes increased blood pressure in germ-free receiving mice [223]. In addition, germ-free mice are resistant to hypertension, vascular dysfunction and have less renal and vascular infiltration of immune cells after infusion of angiotensin II [224]. Both examples of evidence suggest a causal role of the intestinal microbiome in the development of hypertension. A high salt intake in the diet modulates both the composition and the function of the microbiota in rodent models and in humans [225,226]. Several bacterial taxa were observed to be different between hypertensive and normotensive groups: for example, gut microbiome of both hypertensive rats and humans is characterized by an increase in the *Firmicutes/Bacteroidetes* ratio [227]. High salt administration also reduces the prevalence of *Lactobacillus murinus* by increasing the count of splenic pro-inflammatory Th17 cells. Daily administration of *Lactobacillus murinus*, as a probiotic therapy, leads to the reduction of Th17 cells and improves blood pressure in treated rats [228]. Therefore, it can be deduced that the high salt intake and the reduced abundance of species *Lactobacillus* generates a mechanism that causes the interruption of intestinal homeostasis, as well as hypertension. Since the excessive intake of salt causes an alteration that also involves the gut microbiota, it would be desirable, in this condition, to take pre and probiotics, which regulate immune function, improve the intestinal environment, tend to decrease inflammation, increase levels of SCFAs, *Bacteroidetes, Bifidobacterium,* and decrease *Firmicutes* [229,230].

## 3. Discussion 

This review highlights the close connection between hypertension, endothelial dysfunction and gut microbiota. In the first part of the article the meaning of blood pressure and hypertension was developed. Hypertension is the best known risk factor for developing heart failure. In fact, chronic hypertension causes cardiac remodeling within the left ventricle, which culminates in the onset of hypertensive cardiomyopathy and heart failure [231].To date, it is known that there are numerous factors that prevent proper blood pressure control, such as unhealthy lifestyle that includes smoking, alcohol abuse, excess fat and salt in the diet, use of incorrect dosages and/or inappropriate associations of drugs, poor adherence to treatment, overweight and sedentary lifestyles, or prescription of drugs that induce hypertension including nonsteroidal anti-inflammatory drugs, antidepressants, steroids, nasal decongestants and oral contraceptives [232]. Non-pharmacological recommendations for the control of hypertension include weight loss, limited salt and alcohol intake, use of the Dietary Approaches to Stop Hypertension (DASH) diet, high in fruits and vegetables, and intensification of physical activity [233,234]. Although antihypertensive drug treatment is a well-established strategy, hypertension remains poorly controlled worldwide for the following reasons: (1) not all the pathophysiological mechanisms underlying hypertension are fully neutralized by the various classes of pharmacological treatments currently available and (2) the counter-regulatory mechanisms activated by these drugs can reduce their hypotensive effect [235,236,237]. The second part of this review delves into the correlation between hypertension and endothelial dysfunction. In fact, it has been shown that morphological and functional alterations of the endothelium also occur in hypertension, as evidenced by the accumulation of subcutaneous fibrin, by the infiltration of endothelial cells, by alterations in NO-mediated processes and by variations in endothelium-dependent vascular tone. Furthermore, the essential role of the endothelium in the control of inflammation, vascular function, thrombosis and proliferation, makes it particularly involved in hypertension. For this reason, endothelial dysfunction should be considered a central focus for the treatment of hypertension. Several treatments for endothelial dysfunction have been tested for the management of hypertension and, although they have provided promising results, further studies are needed [238,239]. Finally, the third part of this review explores the possible correlation existing between hypertension and the gut microbiota. Despite limited studies, it is now known that not only there is a link between hypertension and endothelial dysfunction but also between endothelial dysfunction and gut microbiota. Indeed, animal models of hypertension have shown concomitant intestinal pathologies and dysbiosis [240,241]. Furthermore, hypertension has been observed to become more pronounced with increased intestinal permeability, fibrosis, decreased calyx cells and villous length in the small intestine [242]. The gut microbiota is capable of releasing certain metabolites which have different effects on endothelial function and blood pressure. Among these, SCFAs, TMAO and uremic toxins may or may not be beneficial for both the endothelium and the control of hypertension [243]. Alongside the great advances made in understanding the important role of gut microbial dysbiosis on hypertension and also other severe cardiovascular diseases in humans and in experimental rodent models, a great deal of attention has turned, in the research, to companion animals such as the dogs and cats. Thanks to the scientific data collected so far, the so-called “gut hypothesis” has been corroborated. According to this hypothesis, the presence of gut dysbiosis could represent early evidence connected to the subsequent onset and progression of hypertension and other cardiovascular diseases when, however, there are no detectable symptoms, thus serving as a potential therapeutic approach for the treatment of cardiovascular and metabolic diseases.

## 4. Conclusions

In conclusion, we suggest that hypertension, endothelial dysfunction, and intestinal microbiota can be considered as the vertices of the same triangle, which are closely related. For the improvement of one of these dysfunctions, it is possible to act on the other two. Despite the data reported in the scientific literature, further confirmatory data would be needed.

## Figures and Tables

**Figure 1 ijms-23-03698-f001:**
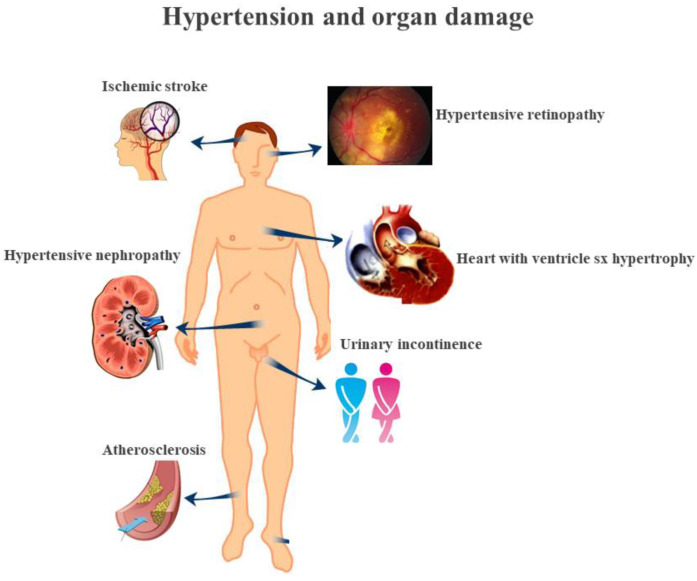
Secondary damage induced by hypertension.

**Figure 2 ijms-23-03698-f002:**
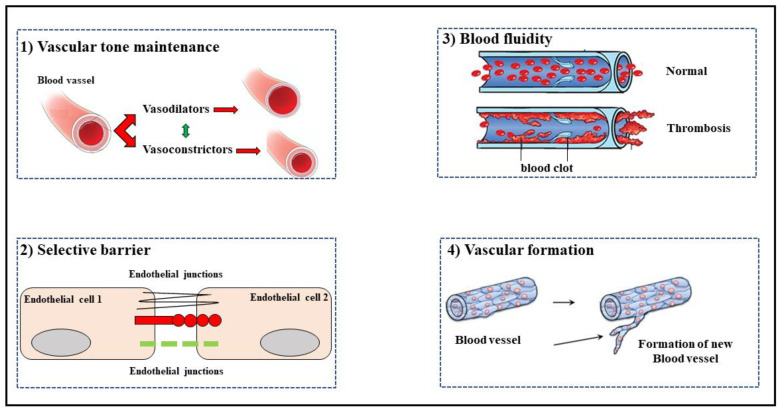
Representation of some endothelium functions.

**Figure 3 ijms-23-03698-f003:**
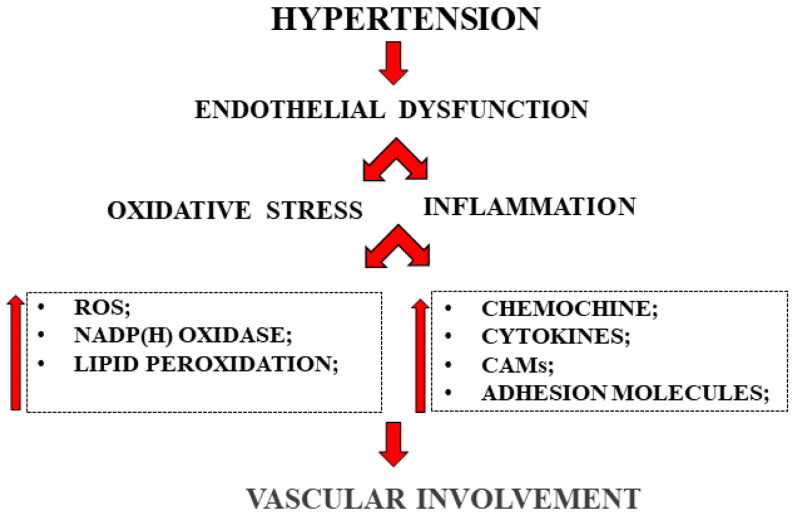
Endothelial involvement in hypertension.

**Figure 4 ijms-23-03698-f004:**
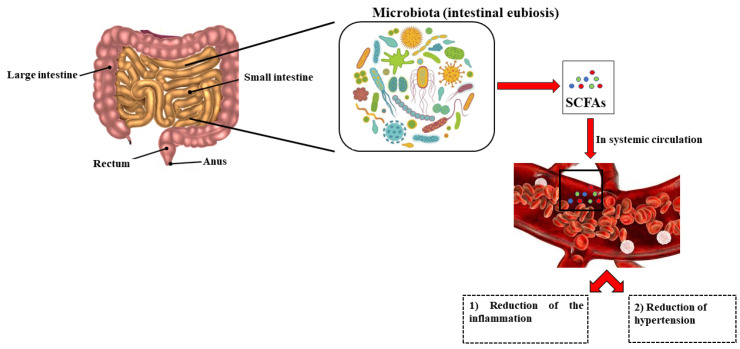
Protective role of the gut microbiota.

**Figure 5 ijms-23-03698-f005:**
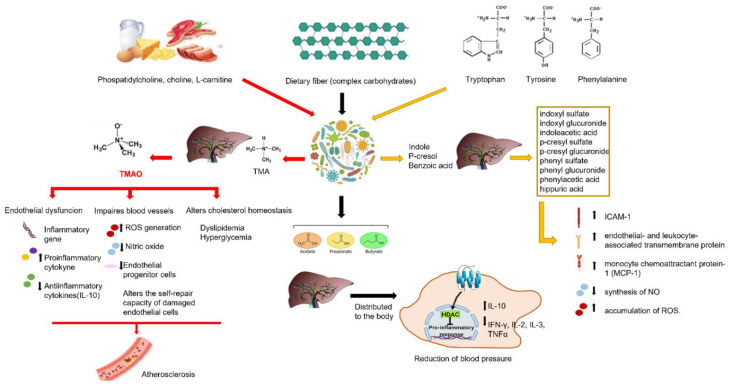
Effects of circulating metabolites on endothelium.

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
