# Peer review of "The Contribution of Gut Microbiota and Endothelial Dysfunction in the Development of Arterial Hypertension in Animal Models and in Humans"

_ijms, 2022, doi:10.3390/ijms23073698_

Round 1
Reviewer 1 Report
This is an interesting and up-to-date review. This review paper provides useful information for researchers in this area. However, this reviewer has several comments as indicated below.
- Introduction is overly too long. Common knowledge (e.g., diagnostic criteria of hypertension) does not need to be referenced. Please shorten the introduction and focus on the main point.
- While the authors mentioned the role of NO in endothelial function and blood pressure regulation, information on EDH, a dominant vasodilator in resistance sized arteries, is missing. In fact, EDH plays a crucial role in endothelial function and blood pressure regulation in many vascular beds and impairment of EDH contributes to endothelial dysfunction in hypertensive animals and humans (Goto et al. Int J Mol Sci 2018). Please include this information in the section “1.2. Endothelial dysfunction and hypertension”.
- With respect to the relationship between G protein-coupled receptors and SCFAs-mediated blood pressure regulation, the authors should cite the paper by Pluznick et al. (Proc Natl Acad Sci USA 2013) and discuss the role of Olfr78 on the SCFAs-mediated blood pressure regulation in the section “2. Gut Microbiota and Hypertension”.
- Several recent studies showed that TMAO impairs endothelial function through inhibiting EDH in certain vascular beds (Matsumoto et al. Biol Pharm Bull 2020, Hamad et al. Biol Pharm Bull 2021). Please add this information in the section “2.1. Gut Microbiota and endothelial dysfunction”.
- Some studies reported that high salt intake alters gut microbial composition, which in turn leads to blood pressure elevation. It would be informative for the readers to refer to this information. This could be briefly discussed in a new paragraph or section.
- There is overlap between the Discussion and Introduction. Please avoid redundancy.
Author Response
This is an interesting and up-to-date review. This review paper provides useful information for researchers in this area. However, this reviewer has several comments as indicated below.
- Introduction is overly too long. Common knowledge (e.g., diagnostic criteria of hypertension) does not need to be referenced. Please shorten the introduction and focus on the main point.
Dear Reviewer Thank you for your valuable suggestions, as mentioned below I’ll point out the changes made:

Reviewer 2 Report
This manuscript is the research about "The contribution of gut microbiota and endothelial dysfunction in the development of arterial hypertension in animal models and in humans" submitted by Jessica Maiuolo * , Cristina Carresi * , Micaela Gliozzi * , Rocco Mollace , Federica Scarano , Miriam Scicchitano , Roberta Macrì , Saverio Nucera , Francesca Bosco , Francesca Oppedisano , Stefano Ruga , Annarita Coppoletta , Lorenza Guarnieri , Antonio Cardamone , Irene Bava , Vincenzo Musolino , Sara Paone , Ernesto Palma , Vincenzo Mollace. this article describes that hypertension is usually characterized by the presence of a chronic increase in systemic blood pressure above the threshold value and is an important risk factor for cardiovascular disease. In addition, the gut microbiota is composed of a highly diverse bacterial population of approximately 1014 bacteria. A balanced intestinal microbiota preserves the digestive and absorbent functions of the intestine, protecting from pathogens and toxic metabolites in the circulation and reducing the onset of various diseases. This review highlights the close connection between hypertension, endothelial dysfunction, and gut microbiota. The manuscript writes well. But some points need to be improved.
The manuscript excludes HPB caused by renal factors, so the GM research data related to the kidney cannot be presented together, which makes "2. Intestinal microbiota and hypertension" in the GM-related references need to improve. In addition, on the issue of HBP caused by "gut microbiota and endothelial dysfunction", the LPS/TLR4 signal transduction pathway, especially
The adjustment mechanism of Toll-Like Receptor Signaling Pathways (TLR4) can be considered for additional columns.
Suggested references (additional) to the following:
- DOI10.1038/nrcardio.2017.120 (IF 32.4)
- DOI10.1016/j.biopha.2021.111334 (IF 6.5)
- https://doi.org/10.3389/fphys.2019.00655 (IF 5.8)
Author Response
Dear Reviewer,
Thank you for your valuable suggestions, as mentioned below I’ll point out the changes made:
In the introduction was added a descriptive part on "hypertension and kidney" in order to justify the portions contained in the following paragraphs. Lines 89-120.
A new paragraph (2.2) deepening the LPS/TLR4 signal transduction path has been added. Lines 604-644.
The papers indicated were very useful and were included in the bibliography.
